# DITTO: QUANTIZATION-AWARE SECURE INFERENCE OF TRANSFORMERS UPON MPC

## ABSTRACT

Due to the rising privacy concerns on sensitive client data and trained models like Transformers, secure multi-party computation (MPC) techniques are employed to enable secure inference despite attendant overhead. Existing works attempt to reduce the overhead using more MPC-friendly non-linear function approximations. However, the integration of quantization widely used in plaintext inference into the MPC domain remains unclear. To bridge this gap, we propose the framework named Ditto to enable more efficient quantization-aware secure Transformer inference. Concretely, we first incorporate an MPC-friendly quantization into Transformer inference and employ a quantization-aware distillation procedure to maintain the model utility. Then, we propose MPC primitives to support the type conversions that are essential in quantization and enable the quantization-aware MPC execution of secure quantized inference. As a result, the computation and communication overhead are reduced, thus enhancing the overall efficiency. We conduct extensive experiments on Bert and GPT2 models to evaluate the performance of Ditto. The results demonstrate that Ditto is about $3.14 \sim 4.40\times$ faster than MPCFormer (ICLR 2023) and $1.44 \sim 2.35\times$ faster than the state-of-the-art work PUMA with negligible utility degradation.

## 1 INTRODUCTION

The recent achievements of pre-trained Transformer Vaswani et al. (2017) models in domains like visual recognition Dosovitskiy et al. (2021); Chen et al. (2021) and natural language processing Devlin et al. (2019); Radford et al. (2019) have led to their widespread adoption for machine learning (ML) inference services. However, despite their increasing popularity, a major concern revolves around data security. In ML services like ChatGPT Brown et al. (2020), the model owner offers an API that receives user prompts as input and generates answers in return. However, this process involves sending user prompts to the server in plaintext, potentially exposing sensitive information. An alternative approach is to employ secure multi-party computation (MPC) techniques Shamir (1979); Yao (1986) based on cryptographic primitives to offer provable security.

However, the huge computation and communication overhead introduced by MPC techniques hinders the application of MPC-based secure Transformer inference. For one thing, non-linear functions like GeLU are frequently invoked, which are extremely expensive in MPC. For another, the overhead is amplified in Transformers due to their large model size. In general, Transformers have millions to billions of parameters and are orders of magnitude larger than traditional ML models. As to the former problem, Chou et al. (2018); Li et al. (2023); Akimoto et al. (2023); Liu & Liu (2023) replace these non-linear functions with MPC-friendly approximations. Regarding the latter, there have been practices in plaintext inference Dettmers et al. (2022); Kim et al. (2021) that leverage mixed-precision quantization to quantize the model parameters to lower bits and employ low-bit integer arithmetic, thus reducing the memory requirement and accelerating the inference. **However, plaintext quantization cannot be trivially incorporated into secure inference upon MPC.** It is worth noting that there are several cross-domain gaps between the worlds of ML and MPC since

- **ML experts** mainly focus on designing delicate quantization methods to improve inference efficiency, which, however, may not be MPC-friendly. The essential type conversions between data types like `INT8` and `FP16` in plaintext quantization are not trivial in MPC. Besides, applying quantization directly may result in a substantial model utility drop.

- **MPC experts** mainly focus on constructing efficient underlying primitives and may not be aware of employing mixed-precision quantization to enhance end-to-end inference efficiency, thus lacking the capability to support quantization-aware secure inference.

Therefore, the problem naturally arises:

*Can we perform quantization-aware secure inference with negligible utility degradation?*

As an answer to the above question, we develop `Ditto`, an *easy-to-deploy* framework for secure and efficient Transformer inference based on a co-design of novel MPC primitives and ML quantization. Specifically, our contributions are as follows:

- **MPC-friendly Quantization-Aware Distillation**. We propose to incorporate *static dyadic quantization* (i.e., from floating-point to fixed-point representation) to avoid CPU-cheap yet MPC-expensive dynamic quantization operations like `clip` and `min/max`. With lower quantization precision, a smaller bitwidth is required, thus reducing the computation and communication overhead in secure inference. Besides, we utilize knowledge distillation to perform *quantization-aware distillation* on pre-trained models to retain the model utility.

- **Secure Quantized Inference Framework.** To the best of our knowledge, `Ditto` is the first framework that supports the MPC execution of quantization-aware secure inference. Concretely, the layer-wise quantized computations are automatically mapped to secure computations over different data types. To do so, we propose MPC primitives to support the interchangeable type conversions. We will open-source the code once accepted.

- **Empirical Evaluations.** We evaluate the performance of `Ditto` in secure inference over several commonly used Transformer models, i.e., Bert and GPT2. The performance is mainly evaluated from two metrics: the model utility and efficiency. The evaluation results indicate that efficiency improvement can be achieved without a significant utility drop. Compared to prior works, `Ditto` is about $3.14 \sim 4.40\times$ faster than MPCFormer Li et al. (2023) and $1.44 \sim 2.35\times$ faster than PUMA Dong et al. (2023).

## 2 RELATED WORK

Transformer-based models have gained significant attention in different scenarios Devlin et al. (2019); Radford et al. (2019); Dosovitskiy et al. (2021), especially after the popularity of large language models Touvron et al. (2023); OpenAI (2023). With their superior performance, Transformers are widely used in machine learning inference services Bommasani et al. (2021); openai. To guarantee the security of input data and trained model parameters, secure inference based on secure multi-party computation (MPC) has been extensively studied.

MPC originates from the Billionaire problem Yao (1986); Shamir (1979) and aims to enable multiple untrusted parties to jointly compute a function while keeping the inputs private. There have been many prior works working on privacy-preserving machine learning (mainly focusing on convolutional neural networks), including **two-party computation (2PC) setting** Mohassel & Zhang (2017); Patra et al. (2021); Huang et al. (2022), **3PC setting** Mohassel & Rindal (2018); Tan et al. (2021); Wagh et al. (2021) and even **4PC setting** Byali et al. (2020); Dalskov et al. (2021). Recent works Li et al. (2023); Hao et al. (2022); Akimoto et al. (2023); Liang et al. (2023); Liu & Liu (2023); Dong et al. (2023) further study the secure inference of more complex Transformer models. These approaches mainly use MPC-friendly approximations for non-linear functions. We take the first step towards leveraging MPC-friendly quantization to enhance the efficiency. Among these works, **3PC in semi-honest and honest-majority setting** Li et al. (2023); Dong et al. (2023) achieves the overall best efficiency. In this work, we also adopt this setting.

## 3 BACKGROUND

In this section, we briefly introduce the Transformer models and the technology of underlying MPC protocols, specifically 2-out-of-3 replicated secret sharing (RSS). Finally, we introduce the quantization methods, along with the fixed-point quantization used in this work.

### 3.1 Transformer and its Variants

Transformer models generally consist of three parts: 1) the `Embedding` module that maps a discrete token to its continuous hidden vector representation; 2) a stack of Transformer `Block`; 3) the last `Prediction` module that maps the hidden vector to task-specific representation. For Transformer `Block`, it typically has `Attention` and `Feed-Forward Network (FFN)` modules.

`Attention` module can be formulated as $\mathsf{Softmax}(\mathsf{Q} \cdot \mathsf{K}^\top + \mathsf{M}) \cdot \mathsf{V}$, where $\mathsf{Q}, \mathsf{K}, \mathsf{V}$ denote the vectors obtained by the matrix multiplication of input activation and three weight matrices, and $\mathsf{M}$ denotes the attention mask. The two widely-used variants, i.e., Bert and GPT, use different masks.

`FFN` module can be formulated as $\mathsf{Linear}(\mathsf{GeLU}(\mathsf{Linear}(x, w_0, b_0)), w_1, b_1)$, where $w_i, b_i$ denote the parameters for $i$-th linear layer. It consists of two linear layers and an activation function GeLU.

### 3.2 Secure Multi-Party Computation

2-out-of-3 replicated secret sharing (RSS) Araki et al. (2016); Mohassel & Rindal (2018), a widely-used MPC technique, runs by splitting a secret value $x$ into several random values (denoted as shares) as $[\![x]\!] = \{x_0, x_1, x_2\}$, s.t., $x = x_0 + x_1 + x_2 \mod 2^\ell$, where $\ell$ denotes the ring size. All the computations are performed over the ring $\mathbb{Z}_2^\ell$. In RSS, the three shares are distributed to three computing parties $\mathcal{P} = \{P_0, P_1, P_2\}$, where $P_i$ holds two shares $\{x_i, x_{i+1}\}$ ($x_3$ corresponds to $x_0$).

In this paper, we use $[\![\cdot]\!]_\ell$ to denote RSS over $\mathbb{Z}_{2^\ell}$. For $\ell \geq 1$ that supports arithmetic operations like $+, -, \cdot$, we denote such type as *arithmetic sharing*. In the case of $\ell = 1$ that only supports boolean operations like bit-wise $\oplus$ and $\wedge$, we refer to this type as *boolean sharing*.

In order to incorporate floating-point arithmetic, which is extensively used in ML, into MPC that operates over a ring, we employ fixed-point quantization to encode floating-point numbers as integers. This approach can be considered as a branch of quantization techniques (refer to Section 3.3).

**Linear Algebra.** Let $a, b, c$ be public constants and $[\![x]\!], [\![y]\!]$ be arithmetic-shared values. $a[\![x]\!] + b[\![y]\!] + c$ only involves addition and multiplication by a public constant. Hence, $[\![ax + by + c]\!]$ can be computed as $(ax_0 + by_0 + c, ax_1 + by_1, ax_2 + by_2)$ locally. While for the multiplication of two shared values, $[\![x \cdot y]\!]$ can be decomposed into $(x_0 + x_1 + x_2) \cdot (y_0 + y_1 + y_2) = \sum_{i=0}^2 z_i = (x_i y_i + x_{i+1} y_i + x_i y_{i+1})$, with $P_i$ computes $z_i$. To obtain $[\![x \cdot y]\!]$, the parties should perform *re-share* $z_i$ Mohassel & Rindal (2018), which requires communication with each other.

**Non-linear Functions.** In addition to linear algebra operations, non-linear functions such as Softmax and Layer Normalization are commonly employed in Transformer inference. To implement these functions, we leverage several underlying MPC computation primitives proposed in prior works Mohassel & Rindal (2018). We omit the descriptions for primitives like comparison, which are used as black boxes, and only present the functionalities of primitives explicitly mentioned in this paper. We refer to Dong et al. (2023); Lu et al. (2020) to construct $\mathsf{Exp}([\![x]\!]) = [\![e^x]\!]$ and $\mathsf{rSqrt}([\![x]\!]) = [\![1/\sqrt{x}]\!]$.

### 3.3 Model Quantization

Quantization Gholami et al. (2021) refers to converting floating-point numbers to low-bit integer representation like 8-bit integer (`INT8`). This can be formulated as $\hat{x} = \mathsf{Int}(\mathsf{Clip}(x, min, max)/S)$, where $min, max$ denote the clipping bound and $S$ denotes the quantization scale. Generally, quantization methods can be divided into two categories: post-training quantization (PTQ) Yao et al. (2022); Dettmers et al. (2022); Frantar et al. (2023) and quantization-aware training (QAT) Kim et al. (2021); Yao et al. (2021). The former PTQ methods allow one-shot quantization while requiring more complex quantization schemes, e.g., more fine-grained token-wise quantization Yao et al. (2022) that uses different $S$ or dynamic quantization Dettmers et al. (2022) that requires computing $min, max$. The latter QAT methods allow for more diverse quantization by quantizing the weights and activations during the training of the model. Hence, cheaper quantization methods like static quantization are feasible despite the cost of *re-training* the model.

Most of the quantization methods utilize floating-point scales to achieve adequate precision. Among these works, *dyadic* quantization Yao et al. (2021); Kim et al. (2021) is a typical class for integer-only quantization, where the scale $S$ is a dyadic number $c/2^f$, $c$ is an integer and $f$ is the precision

bit. In this paper, we employ a modified version called fixed-point quantization ($S = 1/2^f$, with $c = 1$) to accommodate floating-point computations into fixed-point arithmetic, which is crucial in the context of MPC. Fixed-point quantization involves converting floating-point numbers into $\ell$-bit integers using two's complement representation, where the $f$ lower bits represent the fractional part. Mathematically, this can be expressed as $\hat{x} = \text{FXP}_\ell^f(x) = \lfloor x * 2^f \rceil \mod 2^\ell$.

## 4 DESIGN

In this section, we begin by introducing the high-level workflow of Ditto. Then we elaborate on two ingredients in Ditto: 1) the MPC-friendly quantization and distillation of Transformers; 2) the quantization-aware secure inference of the quantized and distilled model upon MPC.

### 4.1 HIGH-LEVEL WORKFLOW

**Setting.** In this paper, we consider the inference scenario, where the model owner provides a trained model $\mathcal{M}$, and the client provides input data $x$ for the inference task. The inference computation can be formulated as $y = \mathcal{M}_\theta(x)$, where $\theta$ denotes the parameters for the model $\mathcal{M}$. The security concern is that both the parameters $\theta$ and input data $x$ are unknown to each other, along with potential attackers. Only the inference result $y$ can be revealed to the client.

Similar to prior works Li et al. (2023); Dong et al. (2023), we consider the secure outsourced 3PC setting. That is, we offload the inference computation to an MPC system consisting of three computing parties $\mathcal{P} = \{P_0, P_1, P_2\}$. The client encrypts $[\![x]\!]$ using RSS and sends the shares to corresponding computing parties. Similarly, the model owner encrypts the model parameters $\theta$ and sends $[\![\theta]\!]$ to $\mathcal{P}$. The computing parties $\mathcal{P}$ then carry out the secure inference and obtain the inference result $[\![y]\!]$. $\mathcal{P}$ then sends all the shares of $y$ to the client, who can reveal the plaintext of $y$.

**Security Model.** In the MPC system, we consider *semi-honest* and *honest-majority* adversary Dong et al. (2023); Tan et al. (2021), where the adversary corrupts no more than half of the computing parties (one exactly in 3PC) and strictly follow the underlying protocol to perform computations but might try to crack sensitive information by collecting and analyzing the messages they receive. We note that output privacy, where the inference outputs can be utilized to infer information like membership Shokri et al. (2017), is beyond the scope.

With the setting and security model in mind, we hereby present the high-level workflow of Ditto in Figure 1. In general, this is a two-step inference scheme via a co-design of ML quantization and efficient MPC computation. The first step (the **upper left** part) is quantizing and distilling the model to a more MPC-friendly version (Section 4.2). This step is performed by the model owner locally using plaintext computation. The second step (the **bottom** part) involves quantization-aware secure inference of the MPC-friendly model obtained from the first step. We design novel MPC primitives to support essential type conversion in quantization (Section 4.3). This step is conducted by the MPC framework, with the model owner and data holder providing inputs.

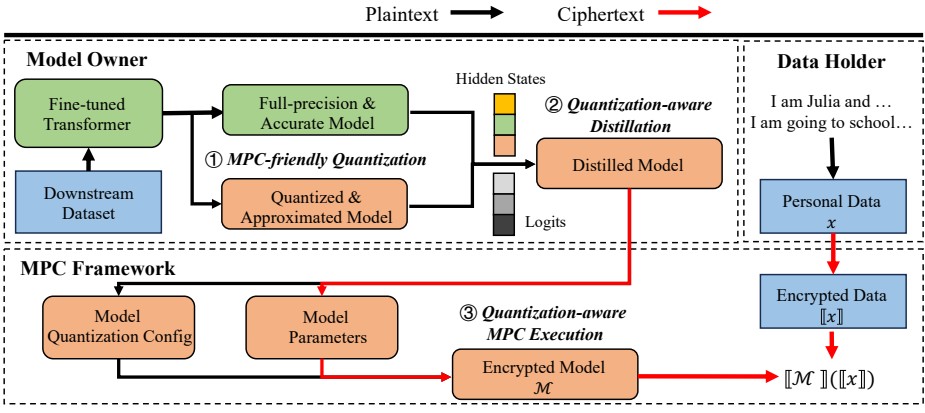

Figure 1: High-level workflow of Ditto.

### 4.2 MPC-FRIENDLY MODEL QUANTIZATION AND DISTILLATION

#### 4.2.1 MPC-FRIENDLY FIXED-POINT MODEL QUANTIZATION

The necessity of model quantization is amplified in secure inference upon MPC. Concretely, the MPC-based secure inference requires communicating messages between the computing parties. The communication size depends on the bitwidth of messages. Therefore, by employing quantization to reduce the bitwidth, the overall communication size can be theoretically reduced, leading to improved inference efficiency. The feasibility of using low-precision quantization is also evidenced in previous works. As observed in Bombari et al. (2022), the neural network models are typically over-parameterized, thus leaving room for reducing the precision and computing with lower bitwidth while maintaining the model utility. The recent success of quantization in large language models Dettmers et al. (2022); Frantar et al. (2022) also prove the feasibility of quantization in more complex and deeper models. Whereas most existing works focus on plaintext inference, there exist several gaps between plaintext quantized inference and secure quantized inference.

***Gap 1: Non-linear functions are different in MPC.*** Most existing plaintext quantization methods use simulated quantization, where the computation of non-linear functions like GeLU and Softmax still operates over floating point arithmetic, thus requiring de-quantizing the quantized integer inputs Bai et al. (2021). Furthermore, the quantization scales are typically stored in floating-point numbers, thus involving complex floating-point arithmetic to convert between different scales. This is not feasible in MPC since all the computations, including non-linear functions and scale conversions, should be computed using integer-only arithmetic over rings.

**Solution.** To facilitate integer-only computations, we employ an **modified dyadic quantization** Kim et al. (2021); Yao et al. (2021) to quantize all the weights and activations into fixed-point numbers, where the quantization scale is in the form of $1/2^f$. In this way, the lower $f$ bits denote the fractional part, and the conversion between different scales can be implemented using left-shift or right-shift (aka. truncation Mohassel & Rindal (2018); Escudero et al. (2020)), which is much cheaper in MPC. Although there is support for secure floating-point computation Rathee et al. (2023), its efficiency is significantly lower compared to secure fixed-point computation.

***Gap 2: Dynamic quantization is expensive in MPC.*** The state-of-the-art plaintext quantization works Dettmers et al. (2022); Frantar et al. (2023) allow the entire inference to be carried out using low-bit integers like `INT8` or even `INT4`. Despite achieving considerable speed up, tailored quantization operations are required, like dynamically computing the min/max/outlier to obtain the scaling factor $S$ and calculating $\lfloor x * S \rceil$ with clipping. Such operations frequently invoke comparisons that are quite expensive in MPC. In this case, directly applying the existing quantization strategy in the secure inference of neural networks is not promising, where the overhead to perform quantization alone even outweighs the communication overhead reduction brought by quantization.

**Solution.** To mitigate this issue, we adopt **static dyadic quantization** to avoid dynamically computing scale in inference. We also adopt layer-wise quantization. That is, we use different quantization scales for different layers. By enabling a smaller quantization scale for linear layers that are not sensitive to precision, we can improve efficiency. While for those non-linear layers like layer normalization, we use a larger scale to avoid a significant accuracy drop. Micikevicius et al. (2017) [1].

***Gap 3: Type conversions are difficult in MPC.*** In plaintext quantization, linear operations are carried out with `INT8` and accumulated in `INT32` in case of overflow. For non-linear functions, the computations operate over `FP32`/`INT32` to achieve adequate precision. Therefore, the end-to-end model inference involves type conversions to avoid overflow and significant precision drop. However, we note that these type conversions between `INT8` and `INT32` are straightforward in plaintext but present a novel challenge in MPC, where a $\ell$-bit integer operates over the ring $\mathbb{Z}_{2^\ell}$ and type conversion involves converting shares among different rings, which cannot be done locally.

**Solution.** To bridge this gap, we propose efficient **type conversion MPC primitives** (Section 4.3.1). Besides, to avoid frequent type conversions, we use uniform low-bitwidth fixed-point encoding ($\text{FXP}_{32}^8$) in intermediate Transformer blocks for linear operations. For the non-linear functions and the last prediction layer, we instead use a higher-bitwidth fixed-point encoding ($\text{FXP}_{64}^{18}$) for the sake

---

[1]Similar precision practices are also used in PyTorch: https://pytorch.org/docs/stable/amp.html#cuda-ops-that-can-autocast-to-float16

of adequate precision. We note that 32/64-bit integers can accommodate the activation range, of which the distribution is illustrated in Appendix A.6, thus avoiding overflow to ensure correctness.

To summarize, we quantize all the weights and activations into fixed-point numbers using layer-wise static quantization with dyadic scales. We provide an illustration of the difference between fixed-point inference and traditional floating-point inference in Appendix A.7. In the following, we describe the computation of non-linear functions using fixed-point only polynomial approximation. The formulated algorithms are presented in Appendix A.2.

**GeLU.** The original GeLU function computes $\mathsf{GeLU}(x) = \frac{x}{2} \cdot (1 + \tanh(\sqrt{2/\pi} \cdot (x + 0.044715 \cdot x^3)))$. To precisely compute GeLU, we first try to use high-order chebyshev polynomial to approximate $\tanh$. However, it requires several multiplication, thus leading to significant overhead. Inspired by Li et al. (2023), we use a quantized polynomial to directly approximate $\mathsf{GeLU}(x) = 0.125x^2 + 0.25x + 0.5$ over $\mathsf{FXP}_{32}^8$. The Quad approximation is worth mentioning as it evaluates a two-order polynomial, allowing it to be computed with lower precision.

**Softmax.** We use Softmax to map the inputs into the range $(0, 1)$ as $\mathsf{Softmax}(x) = \frac{e^{x_i}}{\sum_i e^{x_i}}$. For numerical stability, we first 'normalize' the input by computing $x = x - \max_i(x)$. Since $\max$ does not require a high precision, we compute this part using $\mathsf{FXP}_{32}^8$. For the following exponential and division, we use existing protocols Exp (tailored approximation for Softmax, detailed in Appendix A.10) and Recip Catrina & Saxena (2010). These two functions are computed over $\mathsf{FXP}_{64}^{18}$ to maintain adequate precision. We refrain from using approximations such as 2Relu or 2Quad Li et al. (2023) due to a noticeable drop in accuracy in Li et al. (2023) (in Table 2) and our experiments 5.1.

**Layer Normalization.** Given a vector of $x$ as input, $\mathsf{LayerNorm} = \frac{x - \mu}{\sqrt{\sigma + \epsilon}} \cdot \mathbf{g} + \mathbf{b}$, where $\mu$ and $\sigma$ denote mean and variance, $\mathbf{g}, \mathbf{b}$ denote scale and bias, and $\epsilon$ is a small constant. To avoid significant precision loss, we upcast the inputs and perform the layer normalization with a relatively higher precision $\mathsf{FXP}_{64}^{18}$. The final outputs are downcasted back to $\mathsf{FXP}_{32}^8$ for subsequent computations.

### 4.2.2 QUANTIZATION-AWARE DISTILLATION

Despite the efficiency gain from the above MPC-friendly quantization and approximation, these two steps can cause the precision drop. We illustrate the loss between the outputs from the original model and the quantized and approximated model in Appendix A.5. Consequently, the converted model $\mathcal{M}$ is of low utility. In order to compensate for the error introduced by these two methods, we adopt the methodology of *knowledge distillation* (KD) Jiao et al. (2020); Li et al. (2023).

Without special declaration, we denote the original model as $\mathcal{T}$ and the converted model as $\mathcal{M}$. All the computations in $\mathcal{M}$ use integer-only arithmetic. We leverage layer-wise distillation, considering that we use layer-wise quantization. Concretely, we capture the hidden states of all the Transformer layers from both $\mathcal{T}$ and $\mathcal{M}$ and use the Mean Squared Error loss (MSE) between these two outputs to measure the distillation loss. Furthermore, we use Cross-Entropy loss (CE) between the logits from $\mathcal{M}$ and the target labels as the task-specific loss. Combining these two losses, we obtain the final objective function as $\mathcal{L} = \mathcal{L}_{MSE}(h_{\mathcal{M}}, h_{\mathcal{T}}) + \mathcal{L}_{CE}(logits_{\mathcal{M}}, y)$.

As to the initialization of $\mathcal{M}$, we quantize the weights of $\mathcal{T}$ in the layer-wise granularity without fine-tuning. The quantized weights of low precision serve as the initialed weights of $\mathcal{M}$.

### 4.3 SECURE MODEL INFERENCE UPON MPC

For the end-to-end secure inference in `Ditto`, we rely on existing MPC protocols to perform most of the secure computations. Regarding the type conversions essential in supporting layer-wise quantization, we propose efficient MPC primitives to bridge this gap.

### 4.3.1 TYPE CONVERSION MPC PRIMITIVES

The type conversion can be divided into *upcast* and *downcast*. Upcast refers to converting values from a smaller fixed-point representation to a larger fixed-point representation, while downcast is the opposite. In MPC, type conversions additionally involves share conversions among different rings. We consider convert the input $[\![x]\!]_\ell$ from $\mathsf{FXP}_\ell^f$ to $\mathsf{FXP}_{\ell'}^{f'}$. Due to page limitation, we defer the formulated protocols, along with the correctness analysis and security proof to Appendix A.2~A.4.

**Downcast** ($\ell > \ell', f > f'$). It suffices to a right-shift followed by a modulo operation as $[\![x']\!]_{\ell'} = \mathsf{DownCast}([\![x]\!]_\ell) = x_i \gg (f - f') \mod 2^{\ell'}$ for $i \in \{0, 1, 2\}$. The local right-shift by $(f - f')$ bits first lowers the precision to $2^{f'}$. The subsequent local modulo operation, i.e., dropping the most significant $(\ell - \ell')$ bits, converts the shares to a smaller ring, s.t., $x/2^f = x'/2^{f'}$.

**Upcast** ($\ell < \ell', f < f'$). It suffices to convert $[\![x]\!]$ from $\mathbb{Z}_{2^\ell}$ to $\mathbb{Z}_{2^{\ell'}}$, followed by a left-shift operation. The left-shift can be implemented directly by left-shifting the shares locally. While for the ring conversion, it is not trivial. As shown in Equation 1, there may be potential wrap $w$ of the sum of $x_i$ modulo $2^\ell$, i.e., $w = \lfloor (x_0 + x_1 + x_2)/2^\ell \rfloor$. $w$ cannot be implicitly erased since $\ell < \ell'$ and $w \cdot 2^\ell \mod 2^{\ell'}$ does not equal 0 for sure. However, directly computing $w$ is expensive in MPC.

$$
\begin{aligned}
x \mod 2^\ell &= (x_0 + x_1 + x_2 - w \cdot 2^\ell) \mod 2^\ell \\
&= (x_0 \mod 2^{\ell'}) + (x_1 \mod 2^{\ell'}) + (x_2 \mod 2^{\ell'}) - (w \cdot 2^\ell) \mod 2^{\ell'}
\end{aligned}
\tag{1}
$$

We here take the intuition of mask-and-open that goes as $x = ((x + r) \mod 2^\ell + \hat{w} \cdot 2^\ell - r)$ $\mod 2^{\ell'}$, where $\hat{w} = (x + r) \overset{?}{>} 2^\ell$. The problem now reduces to compute another potential wrap $\hat{w}$ of the sum of $x + r$, which is easier than computing $w$. To facilitate the computation, we add a large bias to the input to make sure the input is positive. Then we can compute $\hat{w} = r_{\ell-1} \wedge \neg y_{\ell-1}$, where $y = x + r \mod 2^\ell$. To finalize the computation, we also need the sharing of $r$ over both $\mathbb{Z}_{2^\ell}$ and $\mathbb{Z}_{2^{\ell'}}$, which can be implemented using DownCast. Detailed construction is shown in Appendix A.2.

### 4.3.2 QUANTIZATION-AWARE MPC EXECUTION

With the underlying MPC primitives ready, we proceed to implement an end-to-end secure quantization-aware inference framework. We build on top of `SecretFlow-SPU` Ma et al. (2023), a framework that supports compiling the front-end models into a privacy-preserving version. To comply with the quantized inference scheme, we make several modifications to `SecretFlow-SPU`. Firstly, we introduce support for dynamic rings in the system runtime. This allows the execution of protocols over different rings, corresponding to underlying data types such as `INT32` or `INT64`. Secondly, we modify the compiler to capture the plaintext variable type and compute type assigned to each operator. During the inference, the variables are automatically converted over different rings by invoking the type conversion MPC primitives. For example, given an operator (e.g., Exp) defined with input type $\mathrm{FXP}_{32}$ and compute type $\mathrm{FXP}_{64}$, an UpCast operator will be automatically invoked to convert the input to $\mathrm{FXP}_{64}$ for subsequent computations.

## 5 EXPERIMENTS

We evaluate `Ditto` mainly from three aspects: 1) model utility (Section 5.1); 2) inference efficiency (Section 5.2); 3) extensive experiments of scalability and ablation studies (Section 5.3).

**Experimental setup.** We implement `Ditto` upon the framework `SecretFlow-SPU` [2] that supports privacy-preserving machine learning. We use pure fixed-point arithmetic during the quantization and distillation procedure, similar to the integer-only arithmetic Kim et al. (2021). We conduct the experiments on one CentOS 8 machine equipped with one AMD Ryzen CPU (32 cores and 3.60GHz) and 256GB of RAM. We consider two network environments: 1) LAN setting with a bandwidth of 5Gbps and 0.4ms round-trip time; 2) WAN setting with a bandwidth of 400Mbps and 40ms round-trip time. We simulate the network environments using the Linux *tc* tool.

**Model architectures and datasets.** We use the pre-trained Bert models and GPT models in Hugging Face Wolf et al. (2020). For Bert, we use Bert-base and Bert-large pre-trained over Book-Corpus Zhu et al. (2015) and English Wikipedia Wikipedia contributors (2004) datasets. For GPT, we use pre-trained GPT2-base and GPT2-medium pre-trained over the Wikitext-103 dataset Merity et al. (2016). We measure the performance of Bert over RTE, CoLA, QQP and QNLI from GLUE benchmarks Wang et al. (2019), and GPT2 performance on the validation set of Wikitext-103. The detailed hyper-parameter choices for fine-tuning and distillation are in Appendix A.8.

**Baselines.** We adopt secure inference upon `SecretFlow-SPU` as the **vanilla baseline**. The ablation models are denoted as `Ditto`$_{w/o\{a\}}$ with quantization, and `Ditto` with both quantization

---

[2] SecretFlow-SPU: https://github.com/secretflow/spu.

and non-linear function approximation. To make a more comprehensive comparison, we compare with two state-of-the-art work MPCFormer Li et al. (2023) and PUMA Dong et al. (2023), which are similar to our setting.

## 5.1 UTILITY EVALUATION

The evaluation of model utility is based on various accuracy metrics for downstream tasks. Concretely, we adopt Accuracy for RTE and QNLI, Matthews correlation for CoLA, F1 score for QQP, and Perplexity for Wikitext-103. In the GLUE benchmark, the input sequence length is set to 128 for Bert-base and Bert-large. For Wikitext-103, the input sequence length is set to 50 for GPT2-base and GPT2-medium. Regarding MPCFormer, we explore two variants: Quad-alone and Quad+2ReLU. For PUMA, the GeLU function is computed using their Poly approximation (cf. Appendix A.9).

The results are provided in Table 1. In general, without approximating Softmax using 2ReLU, Ditto (Quad) achieves similar results to that of MPCFormer (Quad) and slightly lower than the baseline without any quantization or approximation. The utility degradation is negligible on most datasets, except CoLA. The lower utility of both MPCFormer and Ditto on CoLA could be attributed to its smaller size, leading to unstable distillation performance. However, we observe that with 2ReLU approximation, both MPCFormer and Ditto incur noticeable utility drops in Bert tasks. This is in line with the results reported in MPCFormer, thus indicating that Softmax is more sensitive to precision. Regarding PUMA, it is worth noting that it incurs almost no accuracy drop due to the usage of a more accurate Polynomial approximation. However, as demonstrated in the following experiment, this improved accuracy comes at the cost of more communication overhead. To balance between utility and efficiency, we mainly use Quad approximation for GeLU in Ditto.

Table 1: Model utility on GLUE benchmark for Bert and on Wikitext-103 dataset for GPT2.

| Method | Approx. | Bert-base | | | | Bert-large | | | | GPT2-base | GPT2-medium |
|---|---|---|---|---|---|---|---|---|---|---|---|
| | | RTE | CoLA | QQP | QNLI (↑) | RTE | CoLA | QQP | QNLI (↑) | Wikitext-103 (↓) | |
| Baseline | - | 68.59 | 57.06 | 87.96 | 91.62 | 72.56 | 63.09 | 88.52 | 92.58 | 12.25 | 10.60 |
| MPCFormer | Quad | 67.85 | 54.47 | 87.76 | 91.68 | 71.86 | 57.53 | 88.34 | 92.53 | - | - |
| | Quad+2ReLU | 64.30 | 52.75 | 86.95 | 90.76 | 70.29 | 55.53 | 87.64 | 91.85 | - | - |
| PUMA | Poly | 68.47 | 56.96 | 87.95 | 91.48 | 72.56 | 62.60 | 88.50 | 92.55 | 12.25 | 10.49 |
| Ditto$_{w/o\{a\}}$ | - | **67.87** | 54.17 | 87.15 | 91.74 | **72.55** | 56.25 | 88.22 | 92.58 | **12.99** | **10.61** |
| Ditto | Quad | 67.82 | **54.52** | **87.72** | **91.78** | 71.84 | **56.45** | **88.23** | **92.58** | 13.78 | 11.35 |
| | Quad+2ReLU | 63.89 | 52.78 | 86.92 | 87.71 | 71.48 | 51.69 | 87.51 | 87.53 | - | - |

## 5.2 EFFICIENCY EVALUATION

To evaluate the efficiency of secure inference, we measure the end-to-end runtime **in seconds** and the concrete communication size **in GB** in the LAN setting. The experiments are conducted with a batch size of 1. For Bert models, the input sequence length is set to 128, and the output is the classification result. We choose the CoLA task as a representative since other tasks share the same model architecture as CoLA, resulting in similar inference overhead. As for GPT2 models, we generate 1 new token with an input length of 32. We run experiments for MPCFormer [3] and Ditto with and without the Quad approximation of GeLU for a comprehensive comparison.

In Figure 2, the left and right axes represent communication size (**Bar chart**) and runtime (**Line chart**), respectively. On the four models, Ditto (marked in **Red**) generally has the lowest communication size and runtime. One exception is that MPCFormer incurs lower communication size on GPT2-base. This may be because GPT2 models have a larger vocabulary size, thus leading to a much higher communication overhead in the embedding layer for Ditto [4]. Regarding communication size, Ditto$_{w/o\{a\}}$ incurs $1.37 \sim 2.25\times$ lower communication than PUMA, and $1.25 \sim 1.66\times$ than MPCFormer. When combined with Quad approximation, both MPCFormer (Quad) and Ditto incur lower communication than PUMA. Concretely, the communication size of Ditto is $1.28 \sim 1.70\times$ lower than MPCFormer (Quad) and $2.37 \sim 3.43\times$ than PUMA. Owing to the reduction of communication size, Ditto is $3.14 \sim 4.40\times$ and $1.44 \sim 2.35\times$ faster than MPCFormer (Quad) and PUMA, respectively.

---

[3]It is worth noting that MPCFormer is also configured to run on CPU for a fair comparison.

[4]We convert the input token ids to one-hot vectors using MPC, while MPCFormer performs the conversion in the client locally.

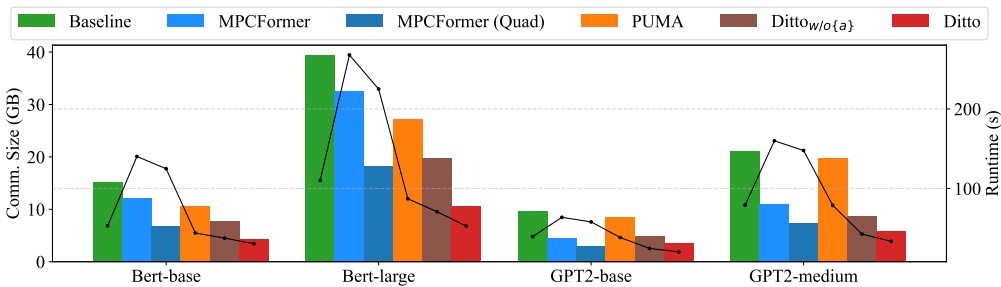

Figure 2: Efficiency evaluation for Bert and GPT2 models.

## 5.3 EXTENSIVE EXPERIMENTS

In this section, we evaluate the inference efficiency with varying input sequence length. The experiments for varying batch size and different network environments are presented in Appendix A.1.

**Varying Input Sequence Length.** The language models typically have conversation sentences as inputs, thus having different input lengths. We hereby conduct experiments with input length $\in \{32, 64, 128, 256\}$ on Bert-base and GPT2-base to make a more comprehensive evaluation.

The results are shown in Table 2 (the speedup numbers are against PUMA). In general, the communication size of Ditto is about $2 \sim 3\times$ lower than the state-of-the-art PUMA. Owing to the communication reduction, Ditto achieves a speedup of about $1.4 \sim 1.8\times$ against PUMA and a speedup of about $2.9 \sim 4.8\times$ against MPCFormer.

Table 2: Inference efficiency on Bert-base and GPT2-base with varying input sequence length.

| Model | Method | #Input Length | | | | | | | |
|-------|--------|---------|------|---------|------|---------|------|---------|------|
| | | 32 | | 64 | | 128 | | 256 | |
| | | Comm. | Time | Comm. | Time | Comm. | Time | Comm. | Time |
| Bert-base | Baseline | 2.79 | 13.57 | 6.24 | 27.32 | 15.12 | 53.17 | 40.65 | 114.83 |
| | MPCFormer | 2.08 | 35.15 | 3.02 | 63.94 | 6.70 | 124.81 | 19.12 | 253.20 |
| | PUMA | 2.16 | 12.97 | 4.65 | 22.59 | 10.59 | 43.98 | 26.07 | 88.70 |
| | Ditto | **0.72** | **7.36** | **1.68** | **14.80** | **4.35** | **30.58** | **12.78** | **62.48** |
| | | 3.00× | 1.76× | 2.77× | 1.53× | 2.43× | 1.44× | 2.04× | 1.42× |
| GPT2-base | Baseline | 8.88 | 37.88 | 12.50 | 52.63 | 22.03 | 84.04 | 47.91 | 157.47 |
| | MPCFormer | **3.05** | 57.57 | **4.01** | 104.97 | **7.73** | 182.20 | 20.24 | 403.85 |
| | PUMA | 8.61 | 35.22 | 11.95 | 48.33 | 17.57 | 73.77 | 33.00 | 131.33 |
| | Ditto | 3.59 | **19.52** | 5.18 | **29.41** | 8.25 | **49.93** | **17.44** | **93.52** |
| | | 2.40× | 1.80× | 2.31× | 1.64× | 2.13× | 1.48× | 1.89× | 1.40× |

**Ablation Studies.** We study the effects of quantization and approximation on Bert models. As shown in Table 3, the quantization with the Quad approximation for GeLU generally results in negligible degradation in utility. The speedup achieved against the vanilla baseline is approximately $1.41 \sim 1.56\times$ with quantization alone and $1.74 \sim 2.09\times$ with the additional GeLU approximation.

Table 3: Ablation studies of Ditto on Bert models.

| Method | Approx. | Bert-base | | | | | Bert-large | | | | |
|--------|---------|-------|-------|-------|-------|---------|-------|-------|-------|-------|---------|
| | | RTE | CoLA | QQP | QNLI | *Speedup* | RTE | CoLA | QQP | QNLI | *Speedup* |
| Baseline | - | 68.59 | 57.06 | 87.96 | 91.62 | - | 72.56 | 63.09 | 88.52 | 92.58 | - |
| Ditto$_{w/o\{a\}}$ | - | 67.87 | 54.17 | 87.15 | 91.74 | 1.41× | 72.55 | 56.25 | 88.22 | 92.58 | 1.56× |
| Ditto | Quad | **67.82** | **54.52** | **87.72** | **91.78** | 1.74× | **71.84** | **56.45** | **88.23** | **92.57** | 2.09× |

## 6 CONCLUSION

In this paper, we propose a framework Ditto to enable secure quantization-aware inference of Transformer models. By incorporating MPC-friendly ML quantization and quantization-aware MPC execution, Ditto reduces the overhead and enhances the inference efficiency compared to prior works. In the future, we plan to investigate adopting more aggressive quantization methods, i.e., using lower bits in secure inference.

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

# A  APPENDIX

## A.1  SUPPLEMENTARY EXPERIMENTS

**Varying Batch Size.**  We evaluate the secure inference with batch size $\in \{1, 2, 4, 8\}$ on Bert-base model. As shown in Table 4, the communication size and runtime increase about linearly to the batch size for all the methods. `Ditto` remains about $1.4\times$ and $4.0\times$ faster than PUMA and MPCFormer, respectively.

Table 4: Inference efficiency on Bert-base with varying batch size. The input length is set to 128.

| Model | Method | #Batch Size | | | | | | | |
|-------|--------|-------|------|-------|------|-------|------|-------|------|
| | | 1 | | 2 | | 4 | | 8 | |
| | | Comm. | Time | Comm. | Time | Comm. | Time | Comm. | Time |
| Bert-base | Baseline | 15.12 | 53.17 | 30.21 | 97.56 | 60.49 | 185.63 | 120.77 | 365.76 |
| | MPCFormer | 6.70 | 124.81 | 11.77 | 227.82 | 21.93 | 432.06 | 42.25 | 839.83 |
| | PUMA | 10.59 | 43.98 | 21.20 | 79.70 | 42.27 | 153.88 | 84.65 | 297.90 |
| | Ditto | **4.35** | **30.58** | **8.68** | **56.29** | **17.35** | **107.52** | **34.96** | **208.63** |
| | | 2.43× | 1.44× | 2.44× | 1.42× | 2.44× | 1.43× | 2.42× | 1.43× |

**Varying Network Environment.**  We evaluate the secure inference under two different network settings, i.e., LAN and WAN. Since secure inference based on MPC is communication-bound, the network status has a significant effect on the efficiency. As shown in Table 5, the runtime increases dramatically in WAN, which is nearly $10\times$ that in LAN. This is because WAN has a smaller bandwidth and higher latency. Compared to PUMA, `Ditto` is still $1.46 \sim 1.53\times$ faster in WAN due to the reduction of communication overhead.

## A.2  FORMULATED PROTOCOL CONSTRUCTIONS

In this section, we present the formulated protocol constructions for the fixed-point computation of GeLU and Softmax functions mentioned in Section 3.3, and the type cast MPC primitives introduced in Section 4.3.1.

The approximation of GeLU function that computes $\mathsf{GeLU}(x) = 0.125x^2 + 0.25x + 0.5$ is presented in Algorithm 1. The Softmax function that computes with different fixed-point representations are shown in Algorithm 2.

Table 5: Inference efficiency of Bert-base and GPT2-base under different network environments.

| Network | Model | Runtime (s) | | | | |
|---------|-------|-------------|-----------|------|-------|-------|
| | | Baseline | MPCFormer | PUMA | Ditto | |
| Bert-base | LAN | 53.17 | 124.81 | 43.98 | **30.58** | 1.44× |
| | WAN | 551.45 | 888.55 | 444.43 | **303.50** | 1.46× |
| GPT2-base | LAN | 39.60 | 57.57 | 38.33 | **20.30** | 1.89× |
| | WAN | 382.98 | 588.14 | 357.65 | **233.32** | 1.53× |

---

**Algorithm 1** Approximated GeLU Protocol

---

**Input:** Fixed-point $x$ over $\texttt{FXP}_\ell^f$; Polynomial coefficients $a, b, c = \{0.125, 0.25, 0.5\}$
**Output:** Fixed-point $y$ over $\texttt{FXP}_\ell^f$;
1: $a_{int} = \lfloor a \cdot 2^f \rceil$, $b_{int} = \lfloor b \cdot 2^f \rceil$, $c_{int} = \lfloor c \cdot 2^f \rceil$
2: $\hat{y} = a_{int} \cdot x/2^f + b_{int}$, $\qquad \triangleright 0.125 \cdot x + 0.25$
3: $y = x \cdot \hat{y}/2^f + c_{int}$, $\qquad \triangleright x \cdot (0.125 \cdot x + 0.25) + 0.5$
4: **return** $y$

---

In the following, the Algorithm 3 and Algorithm 4 depict the construction of downcast and upcast operations in the MPC domain, respectively.

### A.3 CORRECTNESS ANALYSIS

In this section, we analyze the correctness of proposed type cast protocols in Section 4.3.1. The type cast in MPC involves converting shares among rings of different sizes. Consider two rings, $\mathbb{Z}_{2^\ell}$ and $\mathbb{Z}_{2^{\ell'}}$, and let $[\![x]\!]_\ell = \{x_0, x_1, x_2\}$ be the input sharing of $x$ over the first ring. Our goal is to obtain the sharing of $x$ over the second ring, denoted as $[\![x']\!]_{\ell'} = \{x'_0, x'_1, x'_2\}$. We note that $[\![x]\!]_\ell$ and $[\![x]\!]_{\ell'}$ are encoded over $\texttt{FXP}_\ell^f$ and $\texttt{FXP}_{\ell'}^{f'}$, respectively. Hence, we require $x/2^f = x'/2^{f'}$. Note that to ensure correctness, we have the assumption that $x'$ can be represented using $\ell'$ bits, i.e., $x' \in [-2^{\ell'-1}, 2^{\ell'-1} - 1]$.

*Proof.* Based on the relationship between $\{\ell, f\}$ and $\{\ell', f'\}$, we have two cases that correspond to downcast and upcast, respectively.

*Case 1: $\{\ell, f\} > \{\ell', f'\}$ (Downcast).* In Algorithm 3, the input $x$ is firstly right-shifted by $t = f - f'$ bits to lower the precision. The following step is to convert $x/2^{f-f'}$ to the smaller ring $\mathbb{Z}_{2^{\ell'}}$ using modulo operation.

The above steps can be formulated as

$$
\begin{aligned}
x' = x/2^t &= ((x_0 + x_1 + x_2) \mod 2^\ell)/2^t \\
&= x_0/2^t + x_1/2^t + x_2/2^t - w \cdot 2^{\ell-t} + w' \\
&= (x_0/2^t + x_1/2^t + x_2/2^t - w \cdot 2^{\ell-t} + w') \mod 2^{\ell'} \\
&= (x_0/2^t \mod 2^{\ell'}) + (x_1/2^t \mod 2^{\ell'}) + (x_2/2^t \mod 2^{\ell'}) \\
&\quad - (w \cdot 2^{\ell-t} \mod 2^{\ell'}) + w' \mod 2^{\ell'}
\end{aligned}
\tag{2}
$$

where $w' \in \{0, 1, 2\}$ denotes the potential carry bits from the lower $t$ bits. Since we have $\ell - t = 64 - (18 - 8) = 54$, $\ell' = 32$, $w \cdot 2^{\ell-t} \mod 2^{\ell'} = 0$. We can finally get

$$
\begin{aligned}
x' &= (x_0/2^t \mod 2^{\ell'}) + (x_1/2^t \mod 2^{\ell'}) + (x_2/2^t \mod 2^{\ell'}) + w' \\
&= (x'_0 + x'_1 + x'_2 + w') \mod 2^{\ell'}
\end{aligned}
\tag{3}
$$

The probabilistic $w'$ occurs at the lowest significant bit, thus merely having a negligible impact of precision $2^{-f'}$. The correctness of Algorithm 3 thus holds. $\qquad\square$

---

**Algorithm 2** Approximated Softmax Protocol

---

**Input:** Fixed-point $x$ over $\text{FXP}_\ell^f$; $\ell < \ell'$ and $f < f'$
**Output:** Fixed-point $y$ over $\text{FXP}_\ell^f$;
 1: $x = x - \text{Max}(x)$, $\quad \triangleright$ Max computes with precision bit $f$
 2: $\hat{x} = \text{Cast}(x, \text{FXP}_\ell^f, \text{FXP}_{\ell'}^{f'})$, $\quad \triangleright$ from $\text{FXP}_\ell^f$ to $\text{FXP}_{\ell'}^{f'}$
 3: $\hat{x}_{exp} = \text{Exp}(\hat{x})$, $\quad \triangleright$ Exponential computes with precision bit $f'$
 4: $\hat{y} = \hat{x}_{exp}/\text{Sum}(\hat{x}_{exp}, axis = -1)$
 5: $y = \text{Cast}(\hat{y}, \text{FXP}_{\ell'}^{f'}, \text{FXP}_\ell^f)$, $\quad \triangleright$ from $\text{FXP}_{\ell'}^{f'}$ to $\text{FXP}_\ell^f$
 6: **return** $y$

---

**Algorithm 3** Secure DownCast Protocol

---

**Input:** RSS-shared $[\![x]\!]_\ell$ over $\text{FXP}_\ell^f$;
**Output:** RSS-shared $[\![x']\!]_{\ell'}$ over $\text{FXP}_{\ell'}^{f'}$, where $x/2^f = x'/2^{f'}$.
 1: $P_i$ for $i \in \{0, 1, 2\}$ proceed as follows:

$$x'_i = x_i \gg (f - f') \mod 2^{\ell'}$$
$$x'_{i+1} = x_{i+1} \gg (f - f') \mod 2^{\ell'}, \quad \triangleright x/2^{f-f'} \mod 2^{\ell'} = x' \mod 2^{\ell'}$$

 2: **return** $[\![x']\!]_{\ell'} = \{x'_0, x'_1, x'_2\}$.

---

*Case 2:* $\{\ell, f\} < \{\ell', f'\}$ *(Upcast).* The input $x$ is firstly converted to the larger ring $\mathbb{Z}_{2^{\ell'}}$ using Algorithm 4, followed by a left-shifting of $t$ bits. Regarding Algorithm 4, the masking goes as

$$(x + r) \mod 2^\ell = x + r - \hat{w} \cdot 2^\ell \tag{4}$$

where $\hat{w} = (x + r) \overset{?}{>} 2^\ell$. The above equation can be transformed into

$$x \mod 2^{\ell'} = (x + r) \mod 2^\ell - r + \hat{w} \cdot 2^\ell \mod 2^{\ell'} \tag{5}$$

The correctness holds as long as $\hat{w}$ is correct. Recall that we add a bias to ensure that the MSB of $x$ is 0, $x + r$ wraps around $2^\ell$ if and only if the MSB of $r$ (i.e., $r_{\ell-1}$) is 1 and the MSB of $y = x + r$ mod $2^\ell$ (i.e., $y_{\ell-1}$) is 0. Therefore, we can correctly compute the wrap as $\hat{w} = r_{\ell-1} \wedge \neg y_{\ell-1}$. As to the bias, we have a trick that limits the range of $x \in [-2^{\ell-2}, 2^{\ell-2} - 1]$ and choose $2^{\ell-2}$ as the bias. As a result, any input $x = x + r \in [0, 2^{\ell-1} - 1]$ is positive. After the conversion, the bias can be conveniently subtracted to eliminate its influence. The following left-shift operation can be regarded as multiplication by a public constant $2^t$, thus satisfying $x'/2^{f'} = x \cdot 2^t/2^{f'} = x/2^f$.

$\square$

## A.4 Security Proof

**Theorem 1.** *Based on replicated secret sharing, the protocol* DownCast *securely performs the share extension against the semi-honest adversary, with honest-majority assumption.*

*Proof.* The DownCast protocol relies on local right-shift and modulo operations, which are performed individually by each party on the shares they hold. No communication of shares between parties is required for these computations. Owing to the nature of the underlying RSS scheme, each party alone cannot reveal the secret data, proving the overall security of the protocol. $\square$

**Theorem 2.** *Based on replicated secret sharing, the protocol* UpCast *securely performs the share extension against the semi-honest adversary in the* $(\text{PRF}, \text{DownCast})$-*hybrid model, with honest-majority assumption.*

*Proof.* UpCast facilitates the computation by letting $P_2$ locally sample *data-independent* correlated randomness and offload the subsequent computations to the left two parties, i.e., $P_0$ and $P_1$. Recall

---

**Algorithm 4** Secure UpCast Protocol

---

**Input:** RSS-shared $[\![x]\!]_\ell$ over $\mathrm{FXP}_\ell^f$;
**Output:** RSS-shared $[\![x']\!]_{\ell'}$ over $\mathrm{FXP}_{\ell'}^{f'}$, where $x' = x$.
1: $P_2$ proceeds as follows:

    Samples bits $\{r_i\}$ for $i \in [0, \ell-1]$ and computes $r = \sum_{i=0}^{\ell-1} r_i * 2^i$.

    Generates 2-out-of-2 sharing of $r$ and $r_{\ell-1}$ as $\langle r \rangle_{\ell'} = \{r_0, r_1\}$, $\langle r_{\ell-1} \rangle_{\ell'} = \{r_{\ell-1,0}, r_{\ell-1,1}\}$.
    Sends the shares to $P_0$ and $P_1$.    ▷ $P_i$ holds $r_i$ and $r_{\ell-1,i}$, **1 round**.

2: $P_i$ for $i \in \{0,1,2\}$ generate random numbers $z_0, z_2 \in \mathbb{Z}_{2^{\ell'}}$ using PRF:

$$P_2 \text{ and } P_0 \text{ samples } z_0$$
$$P_2 \text{ and } P_1 \text{ samples } z_2$$

3: $P_0$ and $P_1$ obtain $\langle r \rangle_\ell$ by invoking $\mathsf{DownCast}(\langle r \rangle_{\ell'} \ll (f' - f))$,    ▷ using $\langle r \rangle_{\ell'}$ from Step-1

4: $P_0$ and $P_1$ convert $[\![x]\!]_\ell$ to $\langle \hat{x} \rangle_\ell = \{\hat{x}_0, \hat{x}_1\}$ by constructing $\hat{x}_0 = x_1 + x_2$, $\hat{x}_1 = x_0$, where $P_i$ holds $\hat{x}_i$ and $x = \hat{x}$.
5: $P_0$ and $P_1$ executes the following steps:

$$\langle y \rangle_\ell = \langle \hat{x} \rangle_\ell + \langle r \rangle_\ell \text{ and open } y = \mathsf{Reveal}(\langle y \rangle_\ell), \quad ▷ y = \hat{x} + r, \textbf{1 round}$$
$$\langle \hat{w} \rangle_{\ell'} = \langle r_{\ell-1} \rangle_{\ell'} \cdot \neg y_{\ell-1}, \quad ▷ \hat{w} = r_{\ell_1} \cdot \neg y_{\ell-1}$$
$$\langle x' \rangle_{\ell'} = y - \langle r \rangle_{\ell'} + \langle \hat{w} \rangle_{\ell'} \cdot 2^\ell, \quad ▷ x = y - r + \hat{w} \cdot 2^\ell$$

    and outputs $\langle x' \rangle_{\ell'} = \{x'_0, x'_1\}$, s.t., $x'_0 + x'_1 \mod 2^{\ell'} = x$.
6: $P_i$ for $i \in \{0,1\}$ proceed as follows:

    $P_0$ computes $x'_0 = x'_0 - z_0$,   ▷ using random number $z_0$ from Step-2
    $P_1$ computes $x'_1 = x'_1 - z_2$,   ▷ using random number $z_2$ from Step-2
    $P_0$ and $P_1$ exchanges $x'_0$ and $x'_1$   ▷ **1 round**

7: $P_0$ outputs $(z_0, x'_0 + x'_1)$, $P_1$ outputs $(x'_0 + x'_1, z_2)$ and $P_2$ outputs $(z_2, z_0)$.
8: **return** $[\![x']\!]_{\ell'} = \{z_0, x'_0 + x'_1, z_2\}$.

---

that we have honest-majority assumption, $P_2$ cannot collude with either $P_0$ or $P_1$. Hence, although $P_2$ knows the plaintext value of randomness, he cannot reveal the input $x$ without the information of the revealed $y$ in Step-5. The randomness $r$ is a random $\ell$-bit integer. Its sharing over $\mathbb{Z}_{2^{\ell'}}$ is generated by $P_2$ and we let $P_0$ and $P_1$ use DownCast to obtain its sharing over $\mathbb{Z}_{2^\ell}$ in Step-3. As long as the security of DownCast holds, the security of Step-3 holds. Subsequently, the mask-and-open operation computes $y = x + r$ over $\mathbb{Z}_{2^\ell}$. Since $r$ is uniformly sampled over $\mathbb{Z}_{2^\ell}$, and the computation modulos $2^\ell$, information-theoretical security is guaranteed. Despite the information of $y$, both $P_0$ and $P_1$ cannot crack $x$. Regarding the computation of $\hat{w}$, it also merely involves local computations, thus leaking no information to help crack $x$. Finally, in Step-6, the two random values $z_0, z_2$ over $\mathbb{Z}_{2^{\ell'}}$ generated using PRF in Step-2 are used to convert the two-out-of-two sharing of $x'$ into RSS. $P_0$ and $P_1$ use $z_0$ and $z_2$ respectively to mask the shares they hold, while $P_2$ directly take $z_0$ and $z_2$ as his share. Since $z_0$ and $z_2$ are both uniformly sampled over $\mathbb{Z}_{2^{\ell'}}$, the masked shares exchanged between $P_0$ and $P_1$ does not leak any information. Since we assume the pair-wise seeds in PRF are securely distributed to the parties, the security of PRF holds, consequently the security of UpCast.    □

## A.5 ILLUSTRATION OF TRAINING LOSS DURING DISTILLATION

The training curve in Figure 3 depicts the loss between layer-wise outputs of the original model and the model generated by `Ditto` (with quantization and GeLU approximation). It is evident that

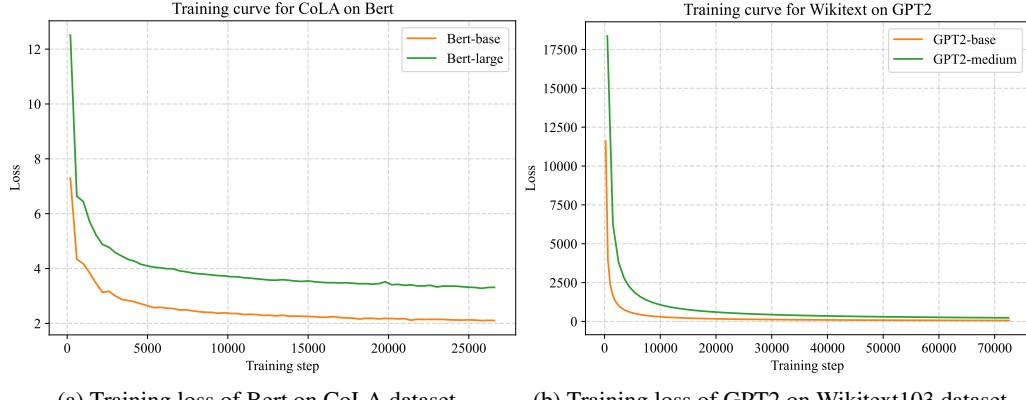

(a) Training loss of Bert on CoLA dataset.  (b) Training loss of GPT2 on Wikitext103 dataset.

Figure 3: Training loss of layer-wise outputs of Bert and GPT2 models.

the model produced by `Ditto` significantly deviates from the original model, with a maximum loss of 12 for Bert and nearly 18000 for GPT2. This substantial divergence indicates that without quantization-aware distillation, the converted model would have a low utility.

## A.6 Illustration of Activation Distribution

In this section, we analyze the activation distribution of Bert and GPT2 models, focusing on the hidden states generated by the intermediate Transformer blocks. As depicted in Figure 4, we observe that the majority of activations in these intermediate layers have absolute values close to zero, with only a small proportion of outliers. For Bert, the outliers fall below 25, while for GPT2, they are below 500. This distribution signifies that the quantization scheme employed in `Ditto` is capable of representing all intermediate values without encountering significant overflows.

## A.7 Illustration of Fixed-point Inference

The difference between fixed-point inference against traditional floating-point inference is illustrated in Figure 5. FP32 denotes float32, and FXP-$\ell, f$ (FXP-$\ell', f'$) represents $\mathrm{FXP}_\ell^f$ ($\mathrm{FXP}_{\ell'}^{f'}$). We have $\ell < \ell'$ and $f < f'$. In floating-point inference (Figure 5a), all the computations are computed using FP32. While in `Ditto` (Figure 5b), all the variables (i.e., activations and weights) in each layer are quantized into fixed-point representation with different precision (marked in orangered). The concrete computations are also carried out using fixed-point arithmetic.

For the linear layers (using $\mathrm{FXP}_\ell^f$), the fixed-point weights and activations serve as the inputs to linear operation, and the outputs are **truncated and clipped** to align the fixed-point representation.

While for the non-linear layers, we first perform **fixed-point conversion** to raise the precision, i.e., from $2^{-f}$ to $2^{-f'}$ for the sake of numerical stability. The non-linear functions are approximated using fixed-point arithmetic (using $\mathrm{FXP}_{\ell'}^{f'}$) that are detailed in Section 4.2.1.

## A.8 Hyper-parameter Choice

**Fine-tuning Configuration.** Regarding the fine-tuning of Bert models for different classification tasks (re-train the last prediction layer), we use a batch size of 32 for Bert-base and 16 for Bert-large. All the inputs are of sequence length 128. We train the models for 3 epochs on RTE, CoLA, QQP and QNLI datasets. We run a grid search with learning rate in [2e-5, 3e-5, 4e-5, 5e-5]. While for GPT2 models, we reuse the trained model parameters from Hugging Face. Concretely, we use pre-trained GPT2-base [5] and GPT2-medium [6] models pre-trained over the Wikitext-103 dataset Merity et al. (2016).

---

[5]GPT2-base on wikitext-103: https://huggingface.co/Graphcore/gpt2-wikitext-103
[6]GPT2-medium on wikitext-103https://huggingface.co/Graphcore/gpt2-medium-wikitext-103

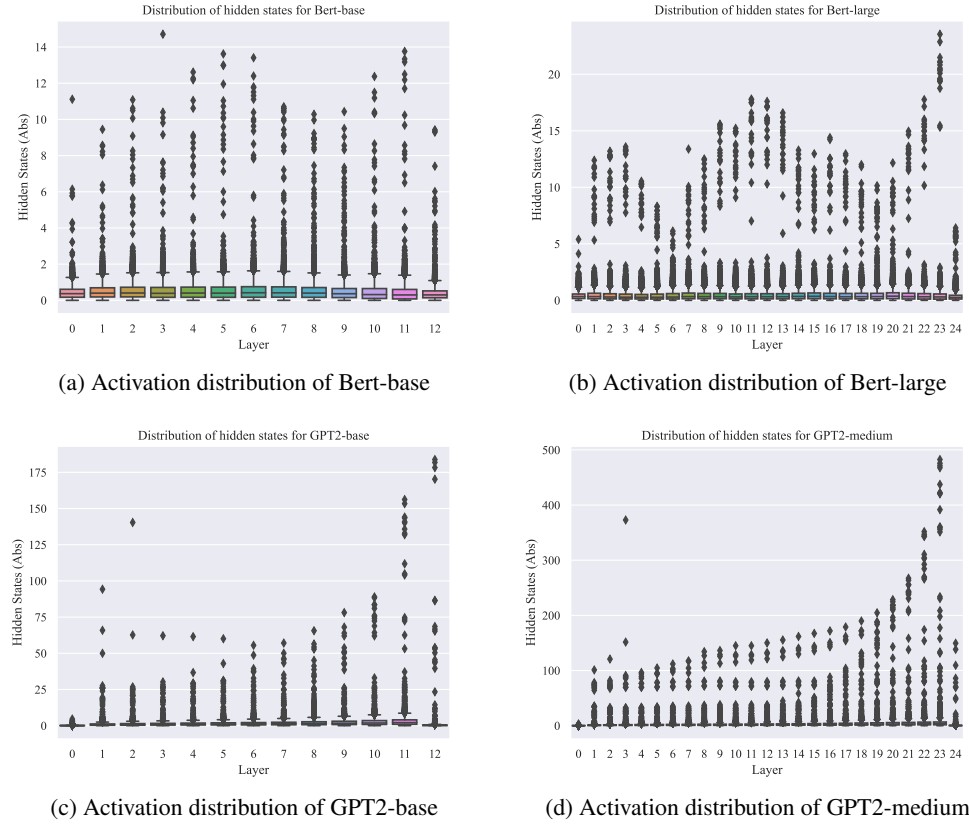

Figure 4: Activation distribution on Bert and GPT2 models.

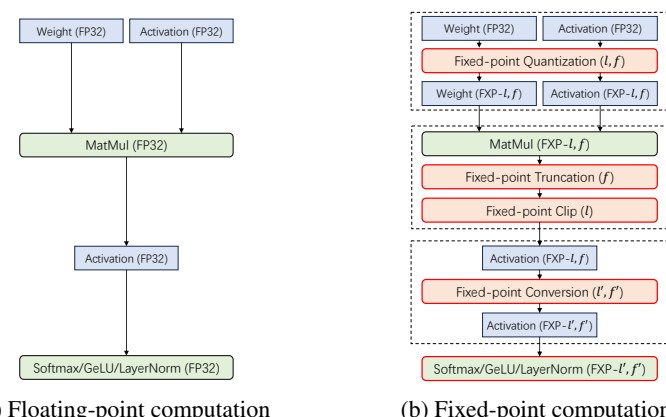

Figure 5: Comparison of fixed-point quantization schemes against original floating-point scheme.

**Distillation Configuration.** The distillation involves two stages: the hidden state distillation and logits distillation, with learning rates of 5e-5 and 1e-5, respectively. In general, we follow the hyper-parameter setting in Li et al. (2023). Regarding Bert models, we train the student model for 50 epochs on RTE, 50 epochs on CoLA, 5 epochs on QQP and 10 epochs on QNLI. All the input sequences are of length 128. For GPT2 models, we train for 1 epoch on Wikitext-103, and the input sequences are of length 50. For the Bert-base model, we use a batch size of 32. While for Bert-large and GPT2 models, we use a batch size of 16 due to GPU memory limitation.

## A.9 POLYNOMIAL APPROXIMATION OF GELU IN PUMA

PUMA Dong et al. (2023) proposed to use a piece-wise approximation of low-degree polynomials for more efficient yet accurate computation of secure GeLU function. In general, the GeLU approximation is split into four splines as follows:

$$
\mathsf{GeLU}(x) = \begin{cases} 0, & x < -4 \\ f_0(x), & -4 \le x < -1.95 \\ f_1(x), & -1.95 \le x \le 3 \\ x, & x > 3 \end{cases}, \tag{6}
$$

where the polynomials $f_0, f_1$ are obtained using numpy.ployfit[7]. The coefficients of the two polynomials are listed below.

$$
\begin{cases} f_0(x) & = -0.011034134030615728x^3 - 0.11807612951181953x^2 \\ & \quad -0.42226581151983866x - 0.5054031199708174 \\ f_1(x) & = 0.0018067462606141187x^6 - 0.037688200365904236x^4 \\ & \quad +0.3603292692789629x^2 + 0.5x + 0.008526321541038084 \end{cases} \tag{7}
$$

## A.10 TAILORED EXPONENTIAL APPROXIMATION

We here describe the tailored exponential approximation for softmax Dong et al. (2023) that is utilized in this work. In general, the exponential function is approximated using a two-segment piecewise function defined in Equation 8. The approximation is specifically designed for the softmax function, which normalizes the input by subtracting the maximum value ($x = x - \max_i(x)$). As a result, the inputs to the exponential function are non-positive, allowing us to employ a Taylor series approximation that maintains precision without significant loss. In our implementation, we set the parameters $T_{\exp}$ to $-14$ and $t$ to $5$ to ensure an efficient and accurate approximation of the exponential function. The specific details and formulas can be found in Dong et al. (2023).

$$
\mathsf{Exp}(x) = \begin{cases} 0, & x < T_{\exp} \\ (1 + \frac{x}{2^t})^{2^t}, & x \in [T_{\exp}, 0]. \end{cases} \tag{8}
$$

---

[7]https://numpy.org/doc/stable/reference/generated/numpy.polyfit.html

