# OpenReview forum: "Ditto: Quantization-Aware Secure Inference of Transformers upon MPC"
_ICLR.cc/2024/Conference — Submitted to ICLR 2024_

### Official Review · Reviewer_iucu · 2023-10-27

**Soundness:** 3 good
**Presentation:** 3 good
**Contribution:** 3 good
**Rating:** 5
**Confidence:** 2

**Summary:**

Author introduce Ditto -- a framework designed to enhance the efficiency of secure inference in Transformer models using multi-party computation (MPC). It incorporates MPC-friendly quantization and a quantization-aware distillation procedure to both reduce computational overhead and maintain model utility. Empirical tests on Bert and GPT2 models show that Ditto significantly outperforms existing solutions, being 3.14 to 4.40 times faster than MPCFormer and 1.44 to 2.35 times faster than PUMA, with negligible loss in utility.

**Strengths:**

* The authors present a solution that addresses multiple bottlenecks in secure multi-party computation (MPC) for Transformer models. For example, challenges like handling non-linear functions and dynamic quantization in an MPC context. They also offer a solution such as modified dyadic quantization and static dyadic quantization for these issues.

* The paper highlights and addresses the often-overlooked disconnect between the expertise in machine learning and multi-party computation. For example, it effectively integrates best practices from MPC-friendly quantization and type-conversion primitives, thereby enhancing end-to-end secure inference efficiency.

* The authors show empirical evidence that their contributions are valid. They compared Ditto against existing state-of-the-art frameworks like MPCFormer and PUMA, the authors make a compelling case for the performance advantages of their approach.

**Weaknesses:**

* The paper acknowledges that both Ditto and MPCFormer exhibit noticeable utility drops in Bert tasks when employing ReLU approximation for Softmax. They offer Quad approximation for GeLU to maintain a balance between utility and efficiency, but this limitation may constrain the applicability of the framework for tasks where such approximations are not tolerable.

* The paper in general is hard to read and require additional proof-reading. I would recommend making the paper to be easier to read by highlighting important concepts, introducing figures that support main results, and describing contributions and future work.

**Questions:**

What are the primary limitations of using more aggressive quantization methods, as mentioned in the future work section, in the context of secure inference? Would it significantly affect model utility, or are there other challenges like security vulnerabilities that need to be addressed?

---

> ### Author Response · Authors · 2023-11-12
>
> We thank the reviewer iucu for the positive feedback on "The paper highlights and addresses the often-overlooked disconnect between the expertise in machine learning and multi-party computation.". We then answer the specific question below.
>
> > They offer Quad approximation for GeLU to maintain a balance between utility and efficiency, but this limitation may constrain the applicability of the framework for tasks where such approximations are not tolerable.
>
> We first note that **the approximation of GeLU is not coupled to Ditto.** In Section 5.2, we present the efficiency of Ditto without approximations, which leads to about $2\times$ communication reduction than prior works. For scenarios where this approximation is not applicable, we can resort to using the standard GeLU or investigating more accurate approximations.
>
> Nevertheless, our comprehensive evaluations, which include models like Bert and GPT, reveal that **the approximated GeLU offers a more favorable tradeoff between utility and efficiency.** As a result, we currently believe that it is feasible to incorporate this approximation into Ditto.
>
> > The paper in general is hard to read and require additional proof-reading.
>
> We thank the detailed review of the paper writing. We will highlight the important concepts in quantization-aware secure inference in the final version.
>
> To summarize, we propose Ditto to bridge the gap between plaintext quantization and MPC-based secure inference. The gap mainly results from two aspects:
>
> 1. (Section 4.3) **Type conversions are difficult in MPC**: MPC encodes **floating-point plaintext** into **fixed-point ciphertexts** (over rings). Typically, the ciphertexts are encoded over a uniform ring of fixed size (like $2^\ell$). In this case, although plaintexts can be quantized into low-bit representation (e.g., from fp16 to int8), the ciphertexts remain the same size (i.e., $\ell$-bit fixed-point numbers). **To incorporate low-bit quantization into MPC, we need to allow ciphertext conversion among different rings. This is why we propose two novel type conversion MPC primitives.** Compared to SOTA type conversion primitives, our protocol incurs much lower communication overhead. Table below lists the theoretical communication complexity against SOTA works (cast from $\mathbb{Z}\_{2^\ell}$ to $\mathbb{Z}\_{2^{\ell'}}$. $\lambda$ is the security parameter, 128 in SIRNN[2]).
>
> |||Round|Size (bits)|
> |:--:|:--:|:--:|:--:|
> |Downcast|SecureML[1]|0|0|
> ||SIRNN[2]|0|0|
> ||Ditto|0|0|
> |Upcast|SIRNN[2]|$\log\ell+2$|$\lambda(\ell+1)+13\ell+\ell'$|
> ||Baccarini et al.[3]|$\log\ell+4$|$2(\ell+\ell')+\log\ell+2$|
> ||Ditto|3|$2\ell+\ell'$|
>
> 2. (Section 4.2) **Dynamic quantization is expensive in MPC**: Standard plaintext quantization requires additional operations like `clamp` and `min/max` during runtime to achieve low-bit quantization (e.g., int8). These operations involve non-linear comparisons that are cheap in plaintext yet quite expensive in MPC. **As a result, we propose an MPC-friendly quantization scheme to improve efficiency.** Besides, we implement the quantization-aware distillation to retain the model utility.
>
> Ditto effectively bridges the aforementioned gaps. **By reducing the size of ciphertexts, we are able to mitigate communication overhead, resulting in improved efficiency against SOTA works.**
>
> > What are the primary limitations of using more aggressive quantization methods, as mentioned in the future work section, in the context of secure inference? Would it significantly affect model utility, or are there other challenges like security vulnerabilities that need to be addressed?
>
> The challenge of employing more aggressive quantization is model utility. In MPC, fixed-point representation, where lower $f$ bits represent the fractional part, is typically used to encode floating-point numbers. **A multiplication doubles the precision to $2f$. If we opt for a more aggressive quantization such as INT16, it would restrict $f$ to be smaller than 8 (since we require $2f < 16$), resulting in a precision lower than $1/2^8$.** As a result, there might be a significant utility drop.
> In the future, we plan to incorporate quantized models into Ditto. This involves taking INT8 weights and INT8 activations as inputs instead of floating-point numbers, allowing us to treat the variables as integers and avoid fixed-point representation. However, achieving this would require additional efforts in terms of re-quantization, which we consider as future work. We believe that incorporating quantized models is an intriguing direction to explore in the realm of MPC-based secure inference.
>
> [1]: Mohassel and Zhang. SecureML: A System for Scalable Privacy-Preserving Machine Learning. IEEE Symposium on Security and Privacy 2017
>
> [2]: Rathee et al. SiRnn: A Math Library for Secure RNN Inference. SP 2021
>
> [3]: Baccarini  et al. Multi-Party Replicated Secret Sharing over a Ring with Applications to Privacy-Preserving Machine Learning. Proc. PETS. 2023

---

### Official Review · Reviewer_MwvF · 2023-11-01

**Soundness:** 3 good
**Presentation:** 3 good
**Contribution:** 3 good
**Rating:** 6
**Confidence:** 4

**Summary:**

This paper proposes MPC primitives to support quantization-aware private inference. Moreover, the authors propose a MPC-friendly quantization-aware distillation to retrain the model utility.

**Strengths:**

1. This paper targets an important problem in private inference.

2. The proposed type conversion protocols are creative solutions to a key challenge in quantization-aware secure inference.

3. Extensive evaluations analyzing efficiency, utility, scalability, and communication costs and latency on factors like sequence length and batch size.

**Weaknesses:**

1. Lack of comparison to the latest related work.

**Questions:**

How would the proposed DITTO be compared with Iron [1]?

[1] Hao, Meng, et al. "Iron: Private inference on transformers." Advances in Neural Information Processing Systems 35 (2022): 15718-15731.

---

> ### Author Response · Authors · 2023-11-12
>
> We thank the reviewer MwvF for the positive feedback on "creative solutions to a key challenge in quantization-aware secure inference.". It is encouraging for us to further narrow the gap between the two entirely different worlds of plaintext quantization and MPC-based secure inference.
>
> We then answer the specific question below.
>
> > Lack of comparison to the latest related work.
>
> Thank you for mentioning Iron. We first present a short comparison against Iron. In general, Ditto is orders of magnitude faster than Iron on Bert (base and large) models with input sequence length of 128.
>
> |       |     Bert-base     |                     |    Bert-large     |                      |
> | :---: | :---------------: | :-----------------: | :---------------: | :------------------: |
> | Iron  | $\approx$ 30 min  |                     | $\approx$ 100 min |                      |
> | Ditto | $\approx$ 0.5 min | $\uparrow 60\times$ |  $\approx$ 1 min  | $\uparrow 100\times$ |
>
> The reason we do not compare Ditto with Iron is because the settings of these two works are not the same. As stated in Section 2, Iron is based on 2PC (2-party computation) setting, while Ditto is 3PC, thus leading to an unfair comparison.
>
> We note that we indeed comprehensively compare with the SOTA works. MPCFormer (ICLR 2023) and PUMA (2023.09) are two latest 3PC works of the same setting as Ditto.

---

> > ### Comment · Reviewer_MwvF · 2023-11-22
> >
> > Thanks for your response. After the second time review, I have two more questions:
> >
> > 1. MPCFormer uses Quad for GeLU and 2Quad for the softmax. Why the authors compare MPCFormer with Quad+2ReLU rather than Quad+2Quad?
> >
> > 2. Are there experimental results on the effects of quantization level on both computational costs and network performance?

---

> > > ### Author Response · Authors · 2023-11-23
> > >
> > > Thank you for your additional questions. Below, we provide specific answers to each of them.
> > >
> > > > MPCFormer uses Quad for GeLU and 2Quad for the softmax. Why the authors compare MPCFormer with Quad+2ReLU rather than Quad+2Quad?
> > >
> > > We want to firstly clarify that MPCFormer evaluates both 2ReLU and 2Quad approximations in their experiments.
> > > Regarding the 2Quad approximation for Softmax, as reported in MPCFormer (Table 4), it causes more utility drops compared to the 2ReLU approximation. Therefore, we can conclude that the utility of 2Quad is worse than that of 2ReLU.
> > > For Ditto, according to Table 1 in Section 5.1, using approximation (e.g. 2ReLU approximation) for Softmax incurs noticeable utility drops.
> > > **Based on the above results, we can safely conclude that Softmax is sensitive to approximations. For the sake of utility, we only adopt the Quad approximation for GeLU to balance efficiency and utility.**
> > >
> > >
> > > > Are there experimental results on the effects of quantization level on both computational costs and network performance?
> > >
> > > In my understanding, the term "quantization level" refers to quantized integers with different bitwidths. Currently, we only have one quantization level: int32+int64. However, it remains as future work to investigate the usage of more aggressive quantization levels, such as int16 or int8. We believe that using a lower quantization bitwidth will result in improvements in both computational costs and network performance. Our experiments have shown that on Bert-base, using lower bitwidths leads to significant improvements.
> > >
> > > |         |  Comm.  |  Time  |
> > > |:-------:|:-------:|:------:|
> > > | 128-bit | 14.12GB | 56.12s |
> > > |  64-bit |  6.24GB | 27.32s |
> > > |  32-bit |  3.20GB | 15.26s |

---

> > > > ### Comment · Reviewer_MwvF · 2023-11-23
> > > >
> > > > Thanks for your reply. It would be good to show Comm. Time as well as the utility for a comprehensive comparison. I will keep my score as it is.

---

> > > > > ### Author Response · Authors · 2023-11-23
> > > > >
> > > > > Thanks for your reply. I hope the above discussion has addressed your concern. Regarding the efficiency (Comm. and Time), you can refer to Section 5.2 and 5.3 for a comprehensive evaluation.

---

### Official Review · Reviewer_sjar · 2023-11-01

**Soundness:** 3 good
**Presentation:** 3 good
**Contribution:** 2 fair
**Rating:** 5
**Confidence:** 4

**Summary:**

This work proposes a framework for quantization-aware secure Transformer inference.

**Strengths:**

+ MPC-friendly Quantization-Aware Distillation.
+ MPC primitives for scale down and scale up.
+ Comparison with SOTA.

**Weaknesses:**

- Distillation is widely used in MPC-based secure inference works.
- It seems limited contributions of MPC protocols.

**Questions:**

1. Does the  Downcast protocol have a probabilistic error? What is the difference compared with the truncation of SecureML?
2. In Upcast, what distribution is $r$ sampled from? How to ensure the input is positive?
3. Could you provide the theoretical or experimental advantages of the proposed Downcast and Upcast protocols compared with SOTA?

---

> ### Author Response · Authors · 2023-11-12
>
> We thank the reviewer sjar for the questions on MPC-side contributions. We are sorry that we defer some detailed descriptions regarding MPC protocols to Appendix due to page limitation.
> We hereby answer the specific questions below.
>
> > Distillation is widely used in MPC-based secure inference works.
>
> We clarify that our main contribution is proposing MPC-friendly quantization and quantization-aware distillation, which effectively bridges the gap between plaintext quantization and MPC-based inference.
> Different from prior works that leverage distillation, we further incorporate MPC-friendly quantization into distillation, which is not trivial. **This involves converting the computations in the student model into fixed-point arithmetic (illustrated in Figure 5, Appendix A.7) and utilizing them as inputs to the distillation process.** With quantization-aware distillation, we can enhance efficiency while maintaining a negligible utility drop.
>
> > It seems limited contributions of MPC protocols.
>
> Due to page limitation, we are unable to provide a detailed description in Section 4.3.1. The **formulated protocol construction**, **correctness analysis** and **security proofs** are presented in Appendix A.2~A.4.
> Besides, we emphasize that the Downcast and UpCast are different from prior works and exhibit better performance, which is the key to quantization-aware secure inference. We believe that the following explanations will help demonstrate the contribution of MPC protocols.
>
> > Does the Downcast protocol have a probabilistic error? What is the difference compared with the truncation of SecureML?
>
> To summarize, there are distinct differences between Downcast and the truncation method in SecureML. **Downcast introduces a probabilistic error solely in the least significant bit (LSB), whereas SecureML introduces probabilistic errors in both the LSB and MSB.** The impact of LSB error is negligible, while the MSB error may cause significant deviations.
>
> As shown in Equation 2, Appendix A.3,
>
> $x'=x/2^t$
>
> $=((x_0+x_1+x_2)\mod 2^\ell)/2^t$
>
> $=x_0/2^t+x_1/2^t+x_2/2^t-w\cdot 2^{\ell-t}+w'$
>
> $=(x_0/2^t+x_1/2^t+x_2/2^t-w\cdot 2^{\ell-t}+w')\mod 2^{\ell'}$
>
> $=(x_0/2^t\mod 2^{\ell'})+(x_1/2^t\mod 2^{\ell'})+(x_2/2^t\mod 2^{\ell'})-(w\cdot 2^{\ell-t}\mod 2^{\ell'})+w'\mod 2^{\ell'}$
>
> there are two potential wraps, i.e., $w$ and $w'$. $w$ leads to the MSB error, while $w'$ leads to the LSB error. In Ditto, since we have $\ell−t=64−(18−8)=54$ and $\ell'=32$, we can get $w \cdot 2^{\ell−t}\mod 2^{\ell'}=0$. That is, we can implicitly erase the MSB error in Ditto. While in SecureML, the potential wrap happens with probability and may lead to significant errors.
>
> > Could you provide the theoretical or experimental advantages of the proposed Downcast and Upcast protocols compared with SOTA?
>
> Consider casting from $\mathbb{Z}\_{2^\ell}$ to $\mathbb{Z}\_{2^{\ell'}}$. $\lambda$ denotes the computational security parameter, typically 128 in SIRNN[2].
>
> The table below demonstrates that Ditto surpasses previous works in terms of the upcast protocol, excelling in both communication round and size. These two factors are crucial in measuring the efficiency of protocols. **Notably, Upcast in Ditto is constant-round (3 rounds), which is much better than SOTA works ($\mathcal{O}(\log\ell)$).**
>
> In terms of downcast, **Ditto incurs no communication like prior works, while avoiding MSB error**, which was a drawback in the SecureML approach.
>
> |||Round|Size (bits)|
> |:------:|:------------------:|:-------------:|:---------------------------------:|
> |Downcast|SecureML[1]|0|0|
> ||SIRNN[2]|0|0|
> ||Ditto|0|0|
> |Upcast|SIRNN[2]| $\log \ell + 2$ | $\lambda (\ell+1) + 13\ell + \ell'$ |
> ||Baccarini et al.[3]| $\log \ell + 4$ |  $2(\ell + \ell') + \log \ell + 2$  |
> ||Ditto|3|$2\ell + \ell'$|
>
> > In Upcast, what distribution is r sampled from?
>
> The randomness $r$ is a random $\ell$-bit integer, uniformly sampled from $\mathbb{Z}_{2^{\ell}}$ to ensure perfect security during the mask-and-open process of $x$. That is, $r \in [-2^{\ell-1}, 2^{\ell-1}-1]$.
>
> > How to ensure the input is positive?
>
> We have a trick that restricts the range of $x\in[-2^{\ell-2},2^{\ell-2}-1]$. As a result, $x+2^{\ell-2}\in[0,2^{\ell-1}-1]$, thus ensuring that the input to Upcast is positive.
> After the conversion, the bias term $2^{\ell-2}$ can be conveniently subtracted to eliminate its influence.
> This approach allows us to achieve performance improvements at the sacrifice of one bit without compromising the overall security of the system.
> The correctness analysis is provided in Appendix A.3.
>
> [1]: Mohassel and Zhang. SecureML: A System for Scalable Privacy-Preserving Machine Learning. IEEE Symposium on Security and Privacy 2017
>
> [2]: Rathee et al. SiRnn: A Math Library for Secure RNN Inference. SP 2021
>
> [3]: Baccarini et al. Multi-Party Replicated Secret Sharing over a Ring with Applications to Privacy-Preserving Machine Learning. Proc. Priv. Enhancing Technol. 2023

---

### Meta-Review · Area_Chair_Emmq · 2023-12-05

**Metareview:**

This paper studies the problem of secure inference for transformer models. It proposes a method for improving the efficiency of inference while preserving utility.
The paper uses the modified dyadic quantization method from prior work in order to quantize all weights and activations into fixed-point numbers. Moreover, it uses a static layer-wise quantization variant from previous work. The paper also proposes efficient type conversion MPC primitives (including upcast and downcast), and employs a particular quantized polynomial approximation to the GeLU function (inspired by prior work). Furthermore, the paper uses the methodology of knowledge distillation, along with an end-to-end secure quantization-aware inference framework from previous works.
The paper includes an experimental evaluation showing that its method (which puts together the previously mentioned ingredients) provides faster inference than the state-of-the-art with roughly the same utility.

The secure inference problem studied in this paper is interesting and well-motivated.
The main downside of the paper is that its novelty is limited. In its present form, it does not meet the bar for acceptance to ICLR.

**Justification For Why Not Higher Score:**

The paper's contributions to are too incremental to meet the bar at ICLR.

**Justification For Why Not Lower Score:**

n/a

---

### Decision · Program_Chairs · 2024-01-16

Reject